# Relationship of Microbial Activity with Soil Properties in Banana Plantations in Venezuela

**Barlin O. Olivares** [1,*] , **Juan C. Rey** [2,3], **Guillermo Perichi** [3] **and Deyanira Lobo** [3]

[1] Programa de Doctorado en Ingeniería Agraria, Alimentaria, Forestal y del Desarrollo Rural Sostenible, Campus Rabanales, Universidad de Córdoba, Carretera Nacional IV, km 396, 14014 Cordoba, Spain

[2] Unidad de Recursos Agroecológicos, Instituto Nacional de Investigaciones Agrícolas (INIA-CENIAP), Av. Universidad vía El Limón, Maracay 02105, Venezuela

[3] Facultad de Agronomía, Universidad Central de Venezuela, Av. Universidad, Maracay 02105, Venezuela

\* Correspondence: barlinolivares@gmail.com or ep2olcab@uco.es

**Abstract:** The present work aims to analyze the relationship of microbial activity with the physicochemical properties of the soil in banana plantations in Venezuela. Six agricultural fields located in two of the main banana production areas of Venezuela were selected. The experimental sites were differentiated with two levels of productivity (high and low) of the "Gran Nain" banana. Ten variables were selected: total free-living nematodes (FLN), bacteriophages, predators, omnivores, Phytonematodes, saturated hydraulic conductivity, total organic carbon, nitrate ($NO_3$), microbial respiration and the variable other fungi. Subsequently, machine learning algorithms were used. First, the Partial Least Squares-Discriminant Analysis (PLS-DA) was applied to find the soil properties that could distinguish the banana productivity levels. Second, the Debiased Sparse Partial Correlation (DSPC) algorithm was applied to obtain the correlation network of the most important variables. The variable free-living nematode predators had a degree of 3 and a betweenness of 4 in the correlation network, followed by $NO_3$. The network shows positive correlations between FLN predators and microbial respiration (r = 1.00; $p$ = 0.014), and $NO_3$ (r = 1.00; $p$ = 0.032). The selected variables are proposed to characterize the soil productivity in bananas and could be used for the management of soil diseases affecting bananas.

**Keywords:** soil microbes; microbial respiration; organic carbon; Phytonematodes; soil; machine learning

## 1. Introduction

Bananas are the most consumed fruits in the world, and one of the most dynamic crops in international trade, considered among the most exported fruits [1]. As a staple food, bananas, including plantains and other types of cooking bananas, contribute to the food security of millions of people in much of the developing world. The banana is rich in carbohydrates (20%), and an important source of Potassium, Vitamin C and Iron [2–4]. According to FAOSTAT [5] Venezuela ranked as the ninth largest banana-producing country in the Americas in 2019 with a production of 650,051 t of bananas from a harvested area of 41,708 ha.

The most recent studies in Venezuela are those of Hernández et al. [6], Delgado et al. [7], González-Pedraza et al. [8], Olivares et al. [9], Martínez et al. [1], Olivares et al. [10,11], González-García et al. [12,13]. The latest research focuses on the properties of the banana soils of Aragua and Trujillo states in Venezuela and emphasizes the importance of microbiological properties in banana productivity.

In agroecosystems, soil organisms are divided into several groups: micro, meso, macro, and megafauna. These organisms are represented by unicellular and microscopic forms, nematodes, insect larvae, earthworms, and arthropods,. They perform the function of transforming all organic and inorganic compounds in the soil, thus having a direct or

indirect influence on properties such as soil porosity, water transport, and the union of particles for the formation of stable soil aggregates [14,15].

In the studies carried out by Delgado et al. [7] and González-García et al. [13] to determine a quality index of banana soils in Venezuela, it was concluded that the biological properties of the soil represented the highest scores in the construction of the quality index. These results establish the importance of considering these variables in research that addresses the quality of banana soils.

In the last decade, the productivity of banana plantations in Venezuela has been affected mainly by the application of high-cost inputs including huge amounts of agrochemicals, and by soil conservation practices, generating a deterioration of the physical, chemical, and biological properties of soil [7,10]. Hence, the importance of evaluating these soil properties is linked to the decomposition of organic matter and the by-products that are generated by their determining role in the soil [14,16].

The characterization, measurement, and analysis of biological diversity in agroecosystems are performed to provide a new and solid knowledge of the different organisms that make their lives in the soil, as well as to obtain indicators that would allow making decisions or recommendations based on the conservation of taxa or areas where the conservation of the agroecosystem is threatened [17–19].

The link to microbial activity is indirectly estimated by determining basal respiration ($O_2$ consumption in the medium or $CO_2$ concentration released) and current productivity; this is fundamental in the search for higher productivity and sustainability in bananas. As noted, most studies rely heavily on soil quality using traditional statistical methods, while the potential of machine learning algorithms such as Debiased Sparse Partial Correlation (DSPC), Partial Least Squares-Discriminant Analysis (PLS-DA), and Random Forest, has been only moderately explored. Various algorithms have been studied; however, to the best of our knowledge, they have not been widely used in banana fields. This is a novel element in our study, in which soil biological properties were explored as promising new soil indicators to assess banana productivity in Venezuelan soils. Our study is a pioneer in the application of algorithms such as DSPC, which is completely new in banana soils. However, beyond its application in the case of a specific crop (banana) and geography (Venezuela), we have developed a scientific logic that is easily transferable to other areas, not only in agriculture but in soil science in general.

In this sense, and motivated by the scarcity of information on the biological quality of soil in the banana areas of Venezuela and its relationship with banana productivity, the objective of this research is to analyze the relationship of microbial activity with the physicochemical properties of the soil in banana plantations in Venezuela and identify the main soil variables responsible for the differentiation between high and low productivity sites, as well as the differences between banana farms. The results of this microbial characterization would reflect the microbiological processes of the organisms from the soil and would be a potential indicator of soil quality.

## 2. Materials and Methods

### 2.1. Description of the Study Areas and Banana Farms

The study area corresponds to six banana farms (Table 1), of which four are in the Depression of the Valencia Lake Basin of Aragua state (PL, SM, PZ, and CH), whose climate is tropical savanna (Aw) with an average annual rainfall of 980.0 mm [20]. The rainfall is seasonal, with five to six rainy months (May-October period). The average annual temperature is 26.2 °C and the average annual relative humidity is 70.0% [21]. The slope ranges from 0–2%. The PL farm is located on the fourth level of the lake terrace, while the SM, PZ, and CH farms are found on alluvial soils, whose texture classes are medium to silty. The predominant soil orders are Mollisols and Inceptisols with characteristics of moderate to good drainage, a neutral to alkaline pH and generally good fertility and organic matter contents above 4.0%.

**Table 1.** Location (latitude, longitude, height), states and bananas planted area of six sites evaluated in Venezuela.

| Farm Code | Latitude | Longitude | Height (masl) | State | Planted Area (ha) | Average Yield (t ha$^{-1}$ year$^{-1}$) |
|---|---|---|---|---|---|---|
| PL | 10°12′20″ N | 67°30′10″ W | 435 | Aragua | 135 | 74.9 |
| BA | 09°29′14″ N | 70°57′05″ W | 16 | Trujillo | 300 | 69.6 |
| KA | 09°28′31″ N | 70°55′46″ W | 17 | Trujillo | 270 | 64.9 |
| SM | 10°12′55″ N | 67°23′42″ W | 502 | Aragua | 11 | 11.1 |
| PZ | 10°11′30″ N | 67°31′04″ W | 514 | Aragua | 20 | 11.4 |
| CH | 10°11′34″ N | 67°31′34″ W | 498 | Aragua | 9 | 12.3 |

The other banana sites (BA and KA) are in the southeast Region of Maracaibo Lake in the Trujillo state, whose climate is tropical savanna (Aw), with annual rainfall amounts of 1094.0 mm with two precipitation peaks, one occurring in April or May and the other in October [22]. The annual average temperature is 27.5 °C and there is an average relative humidity of 78.0%. According to Martinez et al. [1], the terrain is flat with predominantly Entisols soil order, whose drainage is moderate to poor, pH neutral to alkaline, with average organic matter contents of 2.75%; that is, they are soils of moderate fertility.

*2.2. Soil Sampling*

The delimitation of the sampling areas was carried out according to Rosales et al. [23]. To establish the productivity levels in each site, two plots (4.0 ha for PL, BA and KA with four replicated) were identified, and the productivity index (PI) of the "Gran Nain" banana variety was calculated according to the procedure described in Olivares et al. [9]. On the large plantations (≥50 ha, PL, BA and KA), the average yield of high productivity plots was $69.8 \pm 5.0$ t ha$^{-1}$ yr$^{-1}$ and in low productivity plots, it was $59.7 \pm 5.3$ t ha$^{-1}$ yr$^{-1}$. On the other sites (SM, PZ, and CH), the two levels of productivity were identified in lots of 1 ha with two replicated plots in each lot. On the small plantations (<25 ha, SM, PZ and CH) the average yield on small plantations was $11.5 \pm 0.7$ t ha$^{-1}$ yr$^{-1}$ for high productivity plots and 1.6 t ha$^{-1}$ yr$^{-1}$ for those with low productivity. A total of 36 disturbed samples of 2.0 kg of soil were obtained (24 samples for PL, KA, and BA and 12 samples for PZ, CH, and SM) for the determination of the chemical and biological properties of soils by means of soil pits in the first 60 cm depth, following the guidelines suggested in Rosales et al. [23]. The following variables were obtained: total count of populations of bacteria, fungi, and actinomycetes (colony-forming units. g$^{-1}$ soil); microbial respiration (mg 100 g$^{-1}$.10 days$^{-1}$); microbial biomass (mg) and carbon associated with microbial biomass (mg); plant-parasitic nematodes: *Radopholus similis* and *Helicotylenchus multicinctus* (Logarithm (N + 1)); total free-living nematodes (FLN) (number of nematodes in 100 g soil) which includes FLN bacteriophages, predators, and omnivores, Phytonematodes (PN); endophytic fungi in banana root system, namely *Trichoderma*, *Fusarium*; and other fungi. The chemical variables such as nitrate (NO$_3$) included in the analysis were determined according to Rosales et al. [23]. This study considers the same number of undisturbed samples (36) obtained with an Uhland-type sampler (mean depth $29.6 \pm 17.6$ cm) to obtain the saturated hydraulic conductivity (Ks, cm.h$^{-1}$).

*2.3. Statistical Analysis*

Before data analysis, we checked the data integrity. There were no missing values or negative values. The normalization of the soil and microbial variables was carried out using the statistical package in R software version 4.0.2 based on the geometric mean, and a generalized logarithmic transformation using "glog" function in R was performed to make the variables more comparable [24].

Before statistical analysis, a Principal Component Analysis (PCA) was performed as an exploratory analysis to check the presence of outliers and identify patterns in the data. The objective of this application was to summarize the information of many variables (29 in our

study) in a few latent variables, trying to avoid overfitting as new components are added. The PCA allowed us to select only ten variables in our study: total free-living nematodes (FLN); bacteriophages; predators; omnivores; Phytonematodes (PN); saturated hydraulic conductivity (Ks); total organic carbon (C tot); nitrate ($NO_3$); microbial respiration (MR); variable other fungi (which includes non-pathogenic strains of endophytic fungi).

For determining the relative importance of different nematode orders, genera, and microbial variables in the global areas, we used the R circlize package, which provides an implementation of circular layout generation in R as an efficient way for the visualization of microbial information [25].

The Partial Least Squares-Discriminant Analysis (PLS-DA) was applied for the identification of the most important variables in the discrimination of banana productivity sites, which is a supervised method that uses multivariate regression techniques to extract via a linear combination of original variables (X) the information that can predict the class membership (Y). The PLS regression is performed using the plsr function provided by the R pls package [26]. The classification and cross-validation are performed using the corresponding wrapper function offered by the caret package.

The number of optimal components was estimated from cross-validation methods, which calculate the predictive power of the analysis. These methods are based on removing groups of data, calculating a new model from the remaining data, and predicting the values of the removed data. Many iterations of the same process are performed and, from the predicted and observed data, the predictive residual sum of squares (PRESS) is calculated. From the results of all the iterations, the new corresponding $R^2y$ is calculated, and the $Q^2$ that represents the predictive capacity of the model [27]. This parameter is a good indicator to decide which components to include in the model.

The variable importance in projection (VIP) > 1.0, the weighted sum of the PLS-regression coefficient, and corresponding |loading values| > 0.2 in the model were used to identify the key variables [28]. Additionally, the non-parametric Kruskal-Wallis test was applied to obtain the significant variables. Subsequently, the correction was made for the Benjamini-Hochberg multi-test (FDR, false discovery rate) whose established significance level was 95% ($p < 0.05$).

Finally, we applied the Debiased Sparse Partial Correlation algorithm (DSPC), which is based on the recently proposed de-sparsified graphical lasso modeling procedure [29]. Through the graphical result, it is possible to visualize correlation networks where the nodes represent variables under study and the edges represent correlations between them.

## 3. Results and Discussion

### 3.1. Proportion of Free-living Nematodes (FLN) and Phytonematodes (PN) in Banana Sites

According to our results, Phytonematodes predominate in the banana system, with eleven important genera (Figure 1A), such as *Rotylenchulus* (28.5%), *Meloidogyne* (27.6%) and *Helicotylenchus* (24.5%), having the highest proportion. Within the group of FLN, only seven orders were identified, with Rhabditida (60.9%) and Dorylaimida (28.0%) having the highest proportion. Figure 1B shows that no significant differences were found for these nematode variables (data not shown); however, the proportion of Phytonematodes (1.60–3.10 log10) was higher in most of the evaluated banana farms compared to the FLN (1.41–2.48 log10), regardless of the productivity level (Table A2 in Appendix A).

Figure 2A shows the logarithmic representation of bacteria and fungi according to banana productivity levels: high (H) and low (L) in the six evaluated banana sites, indicating that there were no significant differences in representation between the productivity levels; the average representation of bacteria was slightly higher (5.95 ± 0.16 log10) than that of fungi (4.69 ± 0.23 log10) (Table A1 in Appendix A). On the other hand, Figure 2B presents the endophytic fungal populations that colonize the internal tissues of a plant without causing any pathogenic processes. In our case, emphasis was placed on the populations of mutual endophytic fungi such as *Trichoderma*, *Fusarium*, and other fungi because these microorganisms confer protection against pathogen attack and the colonization of

plant organs and tissues. The non-pathogenic strain of *Fusarium* presented an average of $52.71 \pm 20.46$ CFU·g$^{-1}$ soil. It is worth mentioning the high population of *Trichoderma* with $45.00 \pm 21.21$ CFU·g$^{-1}$ soil at the high productivity level of the CH farm, distinguishing it from the rest of the farms. Similarly, the variable "other fungi" was higher in PA (L) (Table A1).

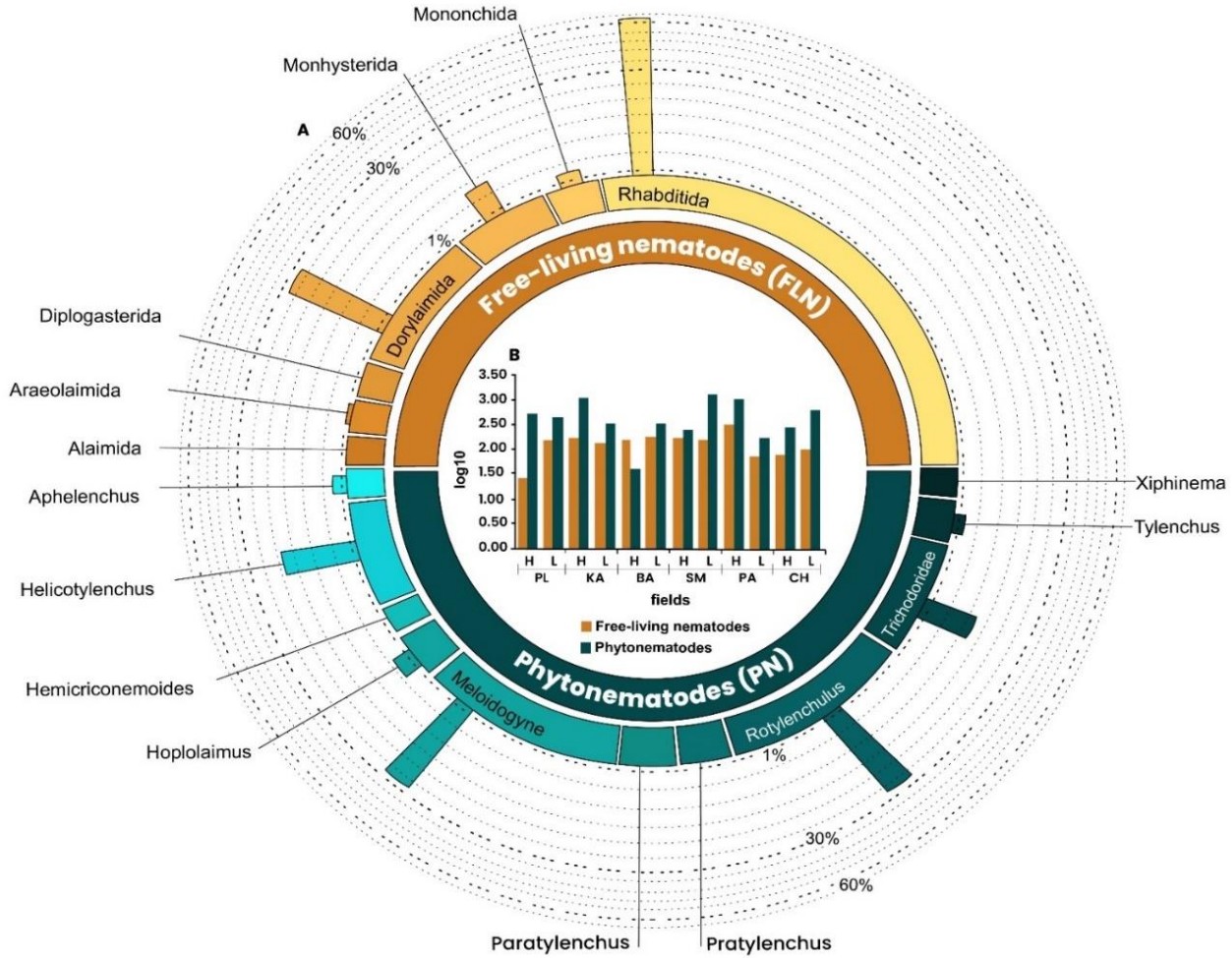

**Figure 1.** (**A**) Proportion of free-living nematodes (FLN) and Phytonematodes (PN) order/genre in the global banana sites. Vertical bars indicate the proportion of species in a group/total number of species in that group×100 (on a logarithmic scale); (**B**) Number of FLN and PN (log10) according to the level of banana productivity: high (H) and Low (L) in the six banana sites: PL, KA, BA, SM, PA, and CH.

Nematodes are the most abundant metazoan and they vary in their sensitivity to environmental disturbances. Free-living and plant-parasitic nematodes are effective ecological indicators, contributing to nutrient cycling and having important roles as primary, secondary, and tertiary consumers in food webs. Tillage, cropping patterns and nutrient management may have strong effects on nematodes, with changes in communities reflecting soil disturbance [30]. In the soils sampled in our study, the bacteria-consuming nematodes were the most representative, followed by the omnivores and lastly the predators of the order Mononchida. The low proportion of this last trophic group belonging to the predatory FLNs could reflect the presence of soil alteration due to intensification and poor soil management in the experimental points evaluated, as pointed out by Olivares et al. [9]; apparently these do not play a significant role as population regulators of the Phytonematodes present. According to Talavera et al. [31], the suppression of phytoparasitic nematodes by predatory nematodes is significant in overly complex soil food webs

and agricultural management leads to a reduction in the suppressive capacity of the soil food web. Our results agree with those of Castilla-Díaz et al. [32] which show the genus c.f. *Dorylaiminae* (Omnivorous) as the most abundant nematode in Colombia. Other studies have found that omnivore and predator nematodes are less abundant in disturbed soils or intensive banana plantations [18].

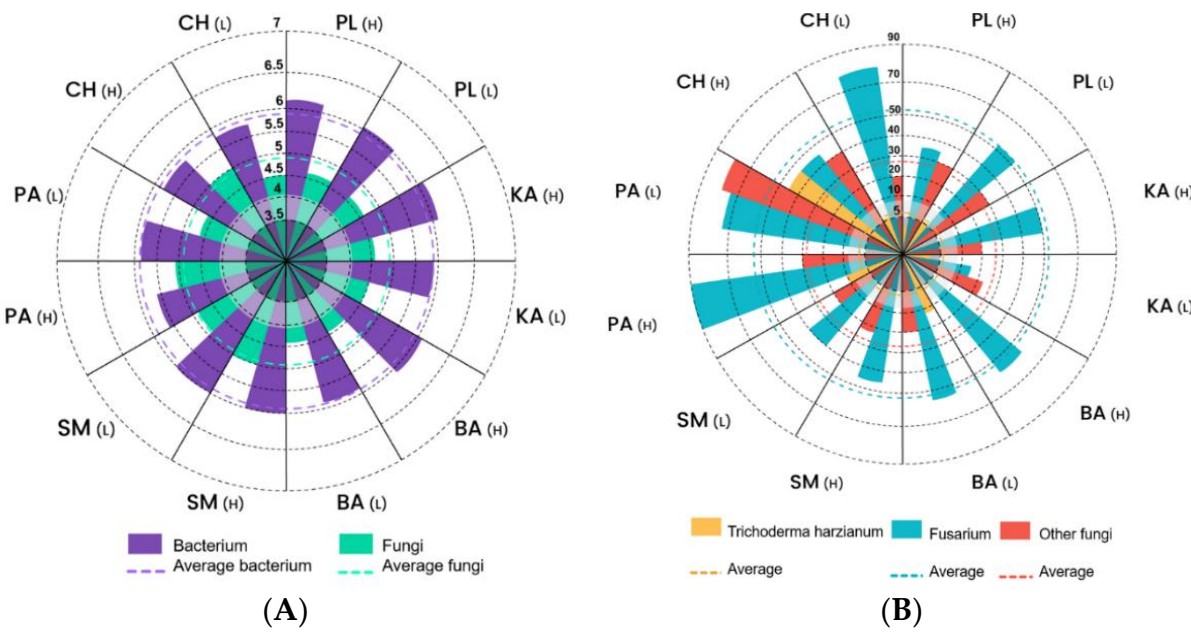

**Figure 2.** Logarithmic representation of: (**A**) bacteria and fungi according to banana productivity levels: high (H) and low (L) in the six evaluated banana sites; (**B**) *Trichoderma harzianum*, *Fusarium* sp. and other fungi in the six evaluated banana sites: PL, KA, BA, SM, PA, and CH.

Our results agree with those of Landi et al. [33] who found that representatives of the order Rhabditida (bacterivores) were the most abundant nematode under different land uses in Italy. Similar results were obtained by Arie-Vonk et al. [34] in the Netherlands with different management practices and soil types. This is because rhabditids, having short life cycles and high reproductive potential, respond more quickly to organic enrichment than other groups of nematodes [35].

Regarding the plant-parasitic nematodes, the most representative genera in this study were: *Rotylenchulus*, *Meloidogyne* and *Helicotylenchus*, respectively. These nematodes are very frequent in banana cultivation, especially in the central region of the country. The most important species are *R. reniformis*, *M. incognita*, *H. dihystera* and *H. multicinctus*. In Aragua State, a combination of *H. multicinctus* and *M. incognita* severely reduces banana yields [36,37].

The study by Davide [38] established that excess K in banana soils increased the penetration and damage of *Meloidogyne* spp., *Helicotylenchus* spp. and *Tylenchorhyndrus* spp. Low levels of potassium 610 mg / l reduced the impact of *R. similis* on banana roots [39]. Likewise, increases in nitrate content in the soil increased the percentage of parasitic nematodes in banana. On the other hand, in banana, a positive correlation was found between the presence of *Meloidogyne* spp. and *H. multicinctus* and the content of silt and sand, as well as a negative correlation with clay [40].

*3.2. Identification of the Most Important Variables with PLS-DA*

The results of the PLS-DA showed the first components of the PLS (accuracy accumulated: 0.61; $R^2y$: 0.50 and $Q^2$: 0.50). Figure 3A shows the scatter charts of the scores and the loading between component 1 and component 2. For the representation, an ellipse with 95% confidence level was constructed from Hotelling's $T^2$ statistic. In Figure 3A, in the plane formed by the 1st and 2nd components, PLS-DA could not distinguish the high

and low productivity levels of bananas (Figure 3A). The first two main components (PC) explained 49.9% of the variables; however, no trends were detected in banana productivity differences. In our case, the optimal number of components is two; new components that are added collect information independent of that of the previous components, so it is to be expected that each time they cover less variability and introduce more noise in the graphs.

According to the criterion of selecting only those variables with |loading values| > 0.2, our results establish that only seven variables meet this condition in component 1, they are: FLN bacterivores (−0.57), FLN predators (−0.52), Ks (−0.42), FLN omnivorous (0.32), C total (0.27), NO$_3$ (−0.25) and MR (−0.21) (Figure 3B). Only corresponding loadings greater than the absolute value of 0.2 are the best option for model interpretation and revealing the most important variables regarding the explanation of the response [41].

On the other hand, two variable importance measures in PLS-DA were used in our study. The first, VIP is a weighted sum of squares of the PLS loadings considering the amount of explained Y-variation in each dimension. VIP scores are calculated for each component. In our study, as the first two components were used to calculate the feature importance, the average of the VIP scores is used (Table 2), with only four variables being the most important: FLN predators, FLN bacterivores, saturated hydraulic conductivity (Ks) and FLN omnivorous. Second, the other important measure is based on the weighted sum of PLS. The weights are a function of the reduction of the sums of squares across the number of PLS components; the average of the feature coefficient is used to indicate the overall coefficient-based importance. The highest coefficients correspond to the variables whose VIP was greater than 1. Table 3 shows the results of the Kruskal Wallis Test and Figure 4 the box graphs where the differences of the significant variables in the evaluated banana farms are evidenced.

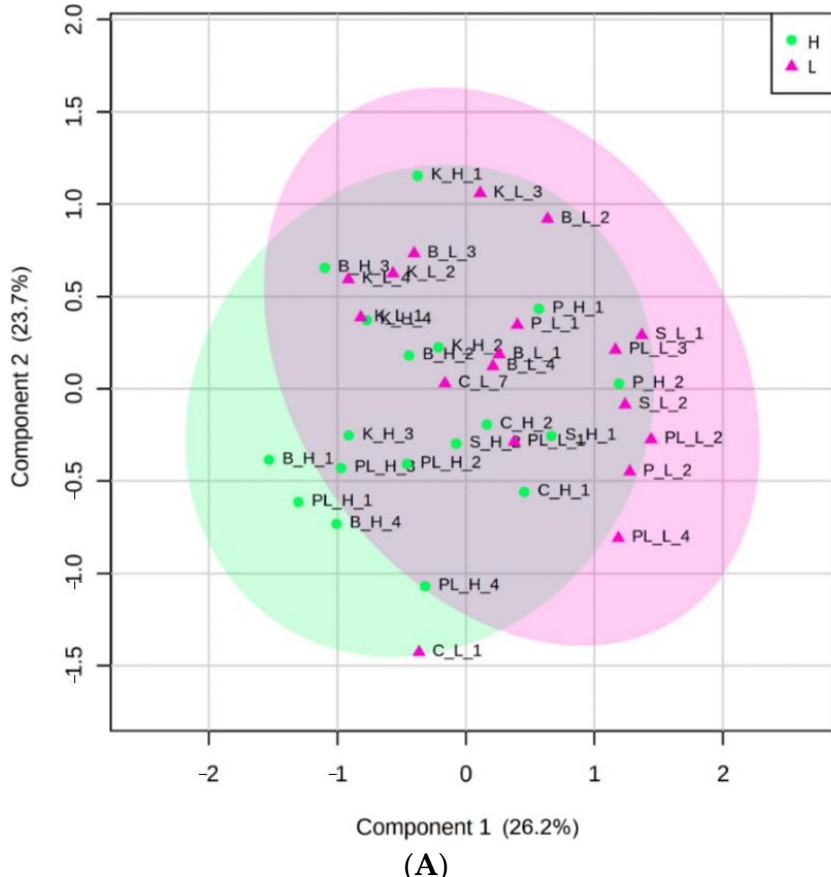

**Figure 3.** *Cont.*

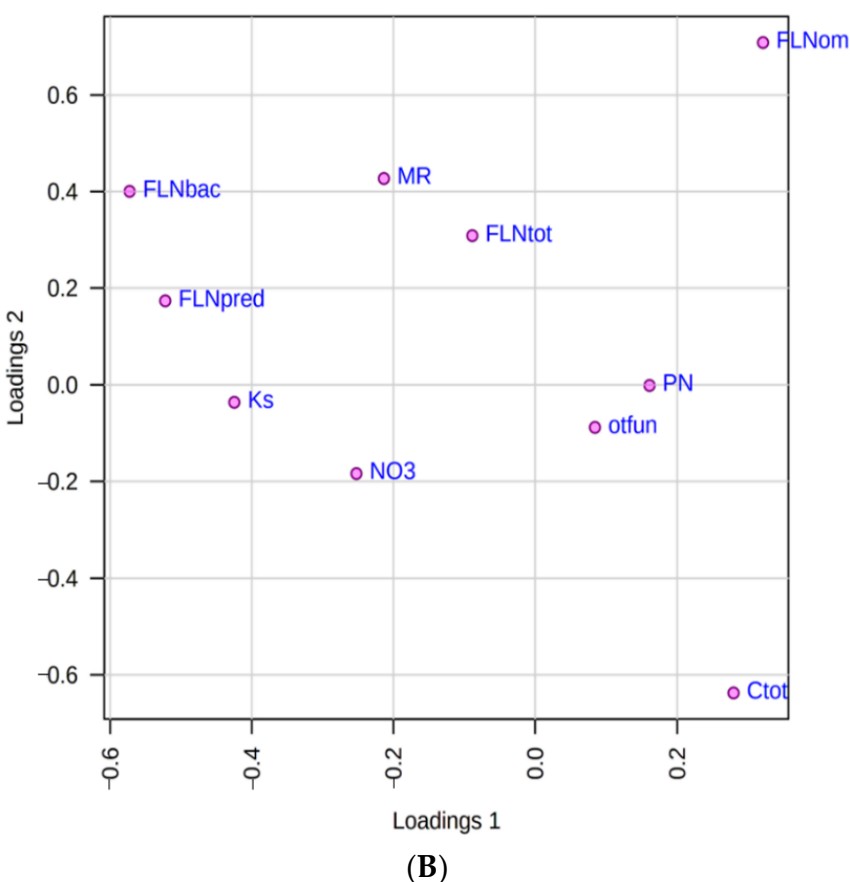

**(B)**

**Figure 3.** (**A**) Scores plot between component 1 and component 2 of the PLS-DA. Note: Data code composed of three digits, the first digit is the site (PL, KA, BA, SM, PA, and CH), followed by the level of productivity: (H) High and (L) low, and finally the repetition number (1–4). The explained variances are shown in brackets. Ellipse T2 de Hotelling (95%); (**B**) Loading plot between component 1 and component 2. (Accuracy accumulated: 0.61; $R^2y$: 0.450 and $Q^2$: 0.50). FLNtot: total free-living nematodes, FLNbac: free-living nematodes bacterivores, FLNpred: free-living nematodes predators, FLNom: free-living nematodes omnivores, Ks: saturated hydraulic conductivity, $NO_3$: nitrate, PN: Phytonematodes, Ctot: total organic carbon, MR: microbial respiration, otfun: other fungi.

**Table 2.** The variable importance in projection (VIP) in componente1, component 2, VIP average and coefficient score.

| Variable | VIP 1 | VIP 2 | VIP Average | Coefficient Score |
|---|---|---|---|---|
| FLN predators | 1.72 | 1.55 | 1.64 | 100.00 |
| FLN bacterivore | 1.48 | 1.40 | 1.44 | 77.58 |
| Ks | 1.41 | 1.27 | 1.34 | 76.43 |
| FLN omnivores | 1.29 | 1.22 | 1.26 | 75.68 |
| $NO_3$ | 0.94 | 0.86 | 0.90 | 47.20 |
| Phytonematodes | 0.31 | 0.37 | 0.34 | 22.31 |
| FLN total | 0.27 | 0.25 | 0.26 | 15.16 |
| Other fungi | 0.20 | 0.20 | 0.20 | 14.95 |
| Total organic carbon (Ctot) | 0.16 | 0.94 | 0.55 | 11.91 |
| Microbial respiration (MR) | 0.08 | 0.76 | 0.42 | 0.00 |

FLN: free-living nematodes; Ks: saturated hydraulic conductivity; $NO_3$: nitrate; VIP: variable importance in projection.

**Table 3.** Important variables by Kruskal Wallis test result: chi-squared, *p*-value (<0.05) and false discovery rate (FDR).

| Variable | Chi-Squared | *p*-Value | $-\log10(p)$ | FDR |
|---|---|---|---|---|
| Microbial respiration (MR) | 22.995 | 0.001 | 34.708 | 0.003 |
| $NO_3$ | 19.706 | 0.001 | 28.481 | 0.007 |
| FLN bacteriophage | 17.479 | 0.003 | 24.347 | 0.010 |
| Total organic carbon (Ctot) | 17.169 | 0.004 | 23.778 | 0.010 |
| FLN total | 15.051 | 0.010 | 19.938 | 0.018 |
| FLN omnivores | 14.85 | 0.011 | 19.577 | 0.018 |

FLN: free-living nematodes; $NO_3$: nitrate; FDR: false discovery rate.

The result demonstrated how soil properties influence the production of a complex and variable system. The Ks, $NO_3$ and microbiological (biotic) properties such as microbial respiration, organic carbon and free-living nematodes were linked to biometric and productive responses in commercial Cavendish banana plantations. This relationship in production can also be related to the response of plants to diseases. However, the variability of the systems must be considered in the eventual implementation of practices to manage soil properties for both production and disease management.

The soil properties obtained through the model show that in some farms the differences between the high and low-productivity sites are very narrow, possibly due to the scarcely differentiated management carried out on the farms. Farm managers do not consider the association between productivity and soil properties, whether physical [11], chemical or biological. This suggests that fertilization is not based essentially on soil or foliar analysis and not on the use of soil conservation practices in the banana sites studied.

Soil management can be a critical factor in PL, BA, and KA banana farms, due to the relationship of the productivity index composed of the number of hands per bunch and the circumference of the mother plant, with the microbiological properties, which can be an indicator of the deterioration of banana production in the evaluated sites.

### 3.3. Debiased Sparse Partial Correlation Algorithm (DSPC)

The results of the deviant sparse partial correlation (DSPC) algorithm that is based on the de-parsified graph lasso modeling procedure [29] are presented. A key assumption underlying our modeling strategy is that the number of true connections between variables is much smaller than the available sample size, that is, the true network of partial correlations between variables is sparse. This assumption is strongly supported by both empirical evidence and theoretical calculations [42–44]. The algorithm DSPC reconstructed a graphical model and provided partial correlation coefficients and *p*-values for each pair of soil and microbial characteristics in the data set (Figure 5). Therefore, DSPC allowed discovering the connectivity between the number of variables and visualizing them as weighted networks where the nodes represent the microbial and soil variables, and the edges represent partial correlation coefficients or the associated *p* values.

Of all the variables involved, the total carbon has a degree of 4 and a betweenness of 7, which represents the importance of the relationships with the other variables of the system. In our case, the microbial activity was measured through the microbial biomass, which is the most active component of the soil; it comprises between 1 and 5% of total carbon and actively participates in the decomposition of organic matter in these banana soils. The variable FLN predators had a degree of 3 and betweenness of 4 in the correlation network, followed by $NO_3$. The network shows positive correlations between MR (r = 1.00; *p* = 0.014), FLN predators (r = 1.00; *p* = 0.032) and $NO_3$.

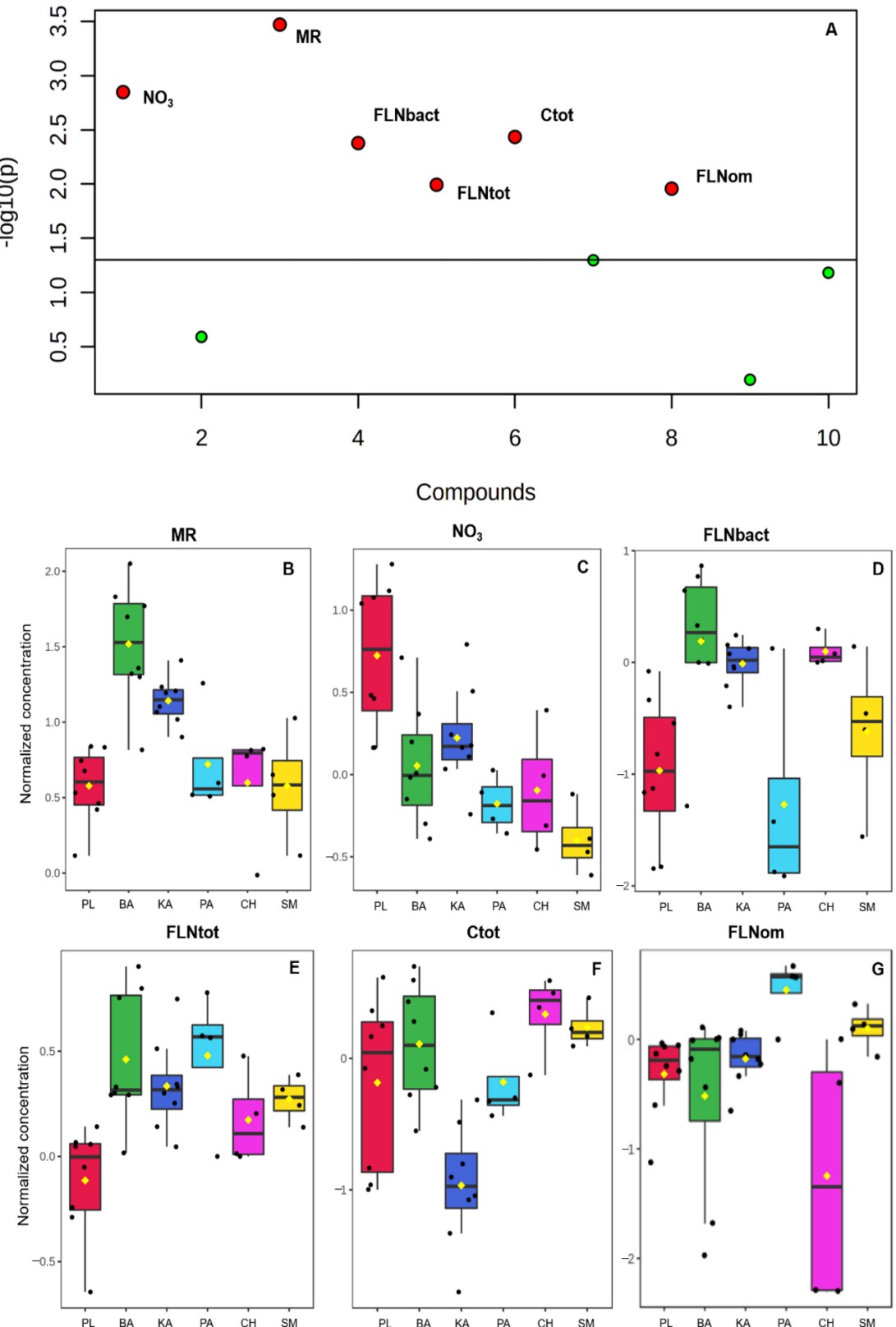

**Figure 4.** (**A**) Important features selected by Kruskal-Wallis's test plot with *p* value threshold 0.05; (**B–G**) Box plots of significant variables in the evaluated banana sites (PL, BA, KA, PA, CH, SM). MR: microbial respiration; $NO_3$: nitrate; FLNbac: free-living nematodes bacterivore; FLNtot: total free-living nematodes; Ctot: total organic carbon; FLNom: free-living nematodes omnivores.

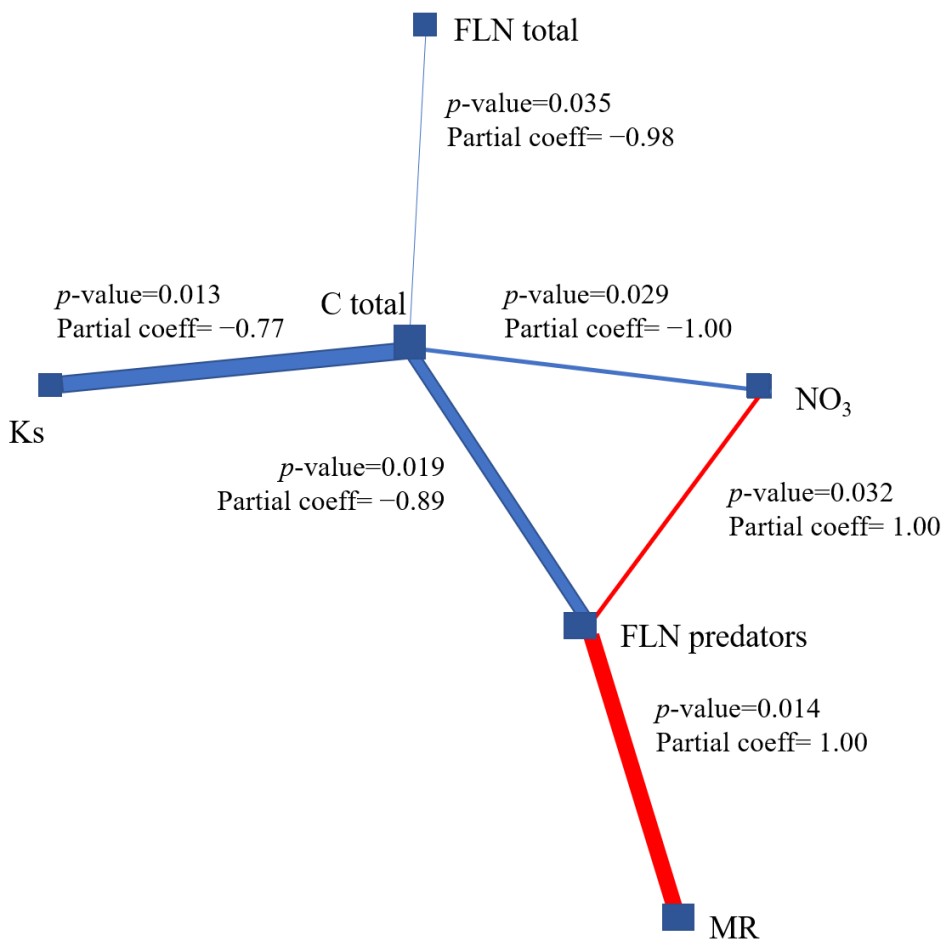

**Figure 5.** Partial correlation network of five evaluated variables. The size of the node indicates the direction of the change. The colored edges had a *p* value <0.05 and the false discovery rate (FDR) adjusted *p* value was <0.5. The red and blue borders show positive and negative correlations. FLNtot: total free-living nematodes; FLNpredators: free-living nematodes predators; Ks: saturated hydraulic conductivity; $NO_3$: nitrate; Ctot: total organic carbon; MR: microbial respiration.

The composition of nematode communities (Phytonematodes and predatory FLNs) is considered a good indicator of alterations such as the release of agrochemicals in the evaluated soils. These changes are correlated with indicators of the functioning of the ecosystem, such as the increase of $NO_3$ in the soil, the decrease in carbon, microbial biomass and the change in the structure of the trophic networks, according to Culman et al. [45]; this is reflected in the negative correlations with Phytonematodes ($r = -0.99$; $p = 0.035$) and predatory FLN ($r = -0.89$; $p = 0.019$) from our study (Figure 5). According to a study of the banana soils of Ecuador [46], the soil physical characteristics (sand, silt, and clay contents, associated with water retention and saturated hydraulic conductivity), 50% of the sampled areas (60 sites) showed a correlation of the weight of roots and with the number of total nematodes.

The study by Ferris et al. [14] revealed that nematodes in the family Tylenchidae, collectively referred to as root associates although some are fungivores, generally do not negatively impact plant growth and respond to increases in organic matter and root weight. Such nematodes contribute to the amplifiable prey pool. Besides providing resources for predator nematodes, they are also prey for micro invertebrate predators of nematodes which, in turn, increases predation pressure on target prey [14,17].

In this regard, [14,17] established that a functional guild of bacterivore and fungivore nematodes participate in organic matter decomposition pathways in the soil. Several

species, often differing in ecological amplitude, may be present at the same time, each contributing to complementarity and continuity of their integral ecological function.

The results of the biological tests in the PL farm indicated a higher total microbial respiration for the high productivity level with respect to the low productivity site, showing that the soils with higher productivity have higher biological activity. In the case of BA and KA, the high-productivity soils presented a greater number of populations of bacteria and fungi, indicating a greater biological activity; this is related to the higher content of microbial carbon in these soils. However, total respiration was lower in high-productivity soils than in low-productivity soils.

In this research, high productivity plots (HH) showed low microbial respiration in all the sites evaluated, except in the PL farm, and in fact, soil organic carbon was not among the soil properties selected to be part of the predictive model of banana productivity based on soil properties. These results are consistent with the experiences of González-Pedraza et al. [8] that indicated the lower amount of microbial carbon and total organic carbon observed in Venezuelan soils with high-productivity plants.

In recent studies, no significant statistical differences were found between lots of different vigor or productivity for microbial carbon, basal soil respiration, and microbial and metabolic ratios [12]. The high microbial activity was closely related to the soil texture, and, in turn, this positively influenced the biometric parameters of the plants. The study by González-García et al. [13] and Rondón et al. [16] shows that the bulk density, content of fine particles, organic matter, and carbon and microbial coefficient, were favorable for the high productivity batches. In general, there were no statistical differences in fungal and bacterial colony-forming units (CFU) between vigor batches.

Higher water retention and higher content of organic matter promote better conditions for the development and activity of soil microorganisms. Results of the effect of these edaphic properties on microorganisms have been reported by Olivares et al. [9] and González-García et al. [13]. The results of Millán et al. [47] showed that the FLN genera found are sensitive to slight changes in soil pH, and show dependence on the porosity and soil moisture. In this regard, [48] showed that there is a significant and important correlation between the carbon of the microbial biomass with the macropores and the Ks (r = 0.84 and 0.59 respectively), that is, the carbon microbial biomass increased when there were good aeration conditions in the soil. In our study, the Ks values did not show significant differences, the highest mean being found in the SM farm with $5.18 \pm 2.70$ cm.h$^{-1}$.

The study by Olivares et al. [9] establishes that commercial banana plantations in the regions of Aragua and Trujillo (Venezuela) were characterized by the intensive use of agrochemicals, generating a considerable reduction in the productivity due to the change and deterioration of the physical, chemical and biological properties of the soil, with the content of Mg, penetration resistance, total microbial respiration, bulk density and omnivorous free-living nematodes being the most determining variables of the quality of banana soil in areas with different levels of productivity. Similarly, [11] indicated that in lacustrine and alluvial banana soils of Venezuela characterized by the change of land use from forest to plantation, the morphological properties of the soil such as biological activity, texture, dry consistency, reaction to HCl and structure type, allow identification of potentially suitable areas for high levels of productivity and long-term sustainability.

The results were compared with the findings of [16], which emphasizes that the high yields of commercial banana farms are associated with a high content of C in stable aggregates, as well as in the more labile fractions of macro-organic matter. These results highlight the importance of the use of less recalcitrant organic fertilizers as a strategy for the sustainable management of banana cultivation.

Likewise, this study represents an important contribution to the knowledge of the banana soils of Venezuela due to the current management of the banana systems in the country. The study can be improved through its systematic application in new locations. It is thus our intention and hopes that other research groups of the international scientific community join this task to produce improved versions.

The research also discusses a series of topics that reflect the thinking and logic of the primary choice of soil variables as the basis of the study. The main use of this research is intended for agricultural extension agents and service provider agents of the banana sector that assist producers of all types. It is also ideal for producers and technical personnel of large and medium-sized farms that design their own production alternatives and undertake problem-solving in Latin America and the Caribbean.

## 4. Conclusions

The methods used in this study to describe the productive capacity or potential of banana soils in Venezuela were based mainly on the study of physical and chemical properties and their relationships with special biological characteristics such as microbial respiration, total proportional free-living nematodes, bacterivores, omnivores, and total organic carbon, which were sufficient to explain the complex interactions of the soil and its rhizosphere.

Nematode populations in the roots and the content of elements in the soil are known to vary both temporally and spatially, which makes this interaction overly complex. The results suggest further studies in two lines: the effect of nutrition on the number of and damage caused by nematodes and their effect on the absorption of nutrients.

The high level of concordance observed in the network is encouraging, as it provides the basis for identifying new connections between soil properties that may represent still undiscovered or poorly studied regulatory interactions of microbial activity in banana soils. The discovery of such novel interactions may lead to a more complete picture of microbial activity in these Venezuelan banana soils.

Practices to increase and maintain soil quality and stimulate microbiological activity in banana soils in Aragua and Trujillo could have a positive effect on agricultural banana production, not only for low-productivity banana lots affected by disease but also for the sustainable use of banana lots of high productivity.

**Author Contributions:** Conceptualization, B.O.O.; methodology, B.O.O., J.C.R. and D.L.; software, B.O.O.; validation, D.L., J.C.R. and G.P.; formal analysis, B.O.O.; investigation, G.P. and B.O.O.; resources,. G.P., D.L.; data curation, J.C.R.; writing—original draft preparation, B.O.O., G.P. and D.L.; writing—review and editing, D.L.; visualization, B.O.O.; supervision, G.P. and D.L.; project administration, J.C.R.; funding acquisition, J.C.R. All authors have read and agreed to the published version of the manuscript.

**Funding:** This research was funded by the project "Technological innovations for the management and improvement of the quality and health of banana soils in Latin America and the Caribbean" financed by FONTAGRO and coordinated by Bioversity International 2007 (before INIBAP).

**Institutional Review Board Statement:** Not applicable.

**Informed Consent Statement:** Not applicable.

**Data Availability Statement:** Not applicable.

**Acknowledgments:** The financial support for international mobility of the Ibero-American Secretary General with Fundación Carolina and Action KA107 of Erasmus+ Program from Agrifood Campus of International Excellence (ceiA3) (2020).

**Conflicts of Interest:** The authors declare no conflict of interest. The funders had no role in the design of the study, in the collection. Analyses, or interpretation of data; in the writing of the manuscript; or in the decision to publish the results.

## Appendix A

**Table A1.** Mean and standard deviation of bacteria, fungi, *Trichoderma harzianum*, *Fusarium* sp. and other fungi according to banana productivity levels: high (H) and low (L) in the six evaluated banana sites: PL, KA, BA, SM, PA, and CH.

| Farm Code | Level | n | Lbact (Log) | Lfung (Log) | Trichoderma CFU·g$^{-1}$ | Fusarium CFU·g$^{-1}$ | Other Fungi CFU·g$^{-1}$ |
|---|---|---|---|---|---|---|---|
| PL | H | 4 | 6.06 ± 0.05 | 4.62 ± 0.05 | 2.50 ± 5.00 | 35.00 ± 31.09 | 30.01 ± 18.26 |
| | L | 4 | 6.13 ± 0.09 | 4.65 ± 0.06 | 5.00 ± 5.77 | 52.50 ± 22.17 | 30.00 ± 14.14 |
| BA | H | 4 | 6.15 ± 0.05 | 4.42 ± 0.15 | 0.00 ± 0.00 | 57.50 ± 30.96 | 0.00 ± 0.00 |
| | L | 4 | 5.81 ± 0.09 | 4.42 ± 0.15 | 12.50 ± 15.00 | 57.50 ± 35.94 | 10.00 ± 14.14 |
| KA | H | 4 | 6.17 ± 0.12 | 4.53 ± 0.15 | 0.00 ± 0.00 | 50.00 ± 14.14 | 20.00 ± 27.08 |
| | L | 4 | 5.92 ± 0.16 | 4.57 ± 0.06 | 2.50 ± 5.00 | 15.00 ± 12.91 | 22.50 ± 17.08 |
| SM | H | 2 | 6.02 ± 0.09 | 5.04 ± 0.62 | 0.00 ± 0.00 | 40.00 ± 56.57 | 20.00 ± 14.14 |
| | L | 2 | 6.02 ± 0.09 | 4.84 ± 0.08 | 0.00 ± 0.00 | 40.00 ± 0.00 | 20.00 ± 14.14 |
| PZ | H | 2 | 5.78 ± 0.11 | 5.04 ± 0.62 | 10.00 ± 14.14 | 90.00 ± 14.14 | 30.00 ± 14.14 |
| | L | 2 | 5.88 ± 0.04 | 4.60 ± 0.00 | 0.00 ± 0.00 | 70.00 ± 42.43 | 80.00 ± 0.10 |
| CH | H | 2 | 5.74 ± 0.06 | 5.04 ± 0.62 | 45.00 ± 21.21 | 45.00 ± 49.50 | 40.00 ± 0.00 |
| | L | 2 | 5.74 ± 0.06 | 4.50 ± 0.28 | 5.00 ± 7.07 | 80.00 ± 28.28 | 20.00 ± 0.00 |

Productivity levels: High (H) and Low (L). CFU = Colony-forming units.

**Table A2.** Mean and standard deviation of number of Free-living nematodes (FLN) and Phytonematodes (PN) (log10) according to the level of banana productivity: high (H) and Low (L) in the six banana sites: PL, KA, BA, SM, PA, and CH.

| Farm Code | Level | n | log10FLN | log10PN |
|---|---|---|---|---|
| PL | H | 4 | 1.41 ± 0.73 | 2.70 ± 1.86 |
| PL | L | 4 | 2.16 ± 1.40 | 2.64 ± 1.67 |
| KA | H | 4 | 2.21 ± 1.00 | 3.03 ± 2.30 |
| KA | L | 4 | 2.12 ± 1.30 | 2.52 ± 1.89 |
| BA | H | 4 | 2.19 ± 1.35 | 1.60 ± 1.09 |
| BA | L | 4 | 2.25 ± 1.58 | 2.54 ± 1.82 |
| SM | H | 2 | 2.20 ± 1.15 | 2.40 ± 2.03 |
| SM | L | 2 | 2.20 ± 1.93 | 3.10 ± 2.39 |
| PA | H | 2 | 2.48 ± 1.63 | 2.99 ± 1.93 |
| PA | L | 2 | 1.85 ± 0.85 | 2.23 ± 2.03 |
| CH | H | 2 | 1.90 ± 0.00 | 2.41 ± 2.2 |
| CH | L | 2 | 2.00 ± 1.15 | 2.77 ± 1.8 |

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
