# Peer review of "Relationship of Microbial Activity with Soil Properties in Banana Plantations in Venezuela"

_sustainability, doi:10.3390/su142013531_

Round 1
Reviewer 1 Report
I have reviewed the manuscript entitled " Relationship of microbial activity with soil properties in banana plantations in Venezuela". The authors study relationship between microbial communities, soil physicochemical properties in different cropland area. The manuscript is generally well written. However, the some statistical analysis is problematic and the lack of details in the results section.
L70: should clarify which kind of microbial activity is discussed in the manuscript.
Fig 1b: raw data points and stdev is missing
L195-197: the authors should do ANOVA or similar analysis to compare the nematode variables
Fig 2: The logarithmic looks good but the author needs to show the stdev and number of replicates
L275: why the loading values have to be higher than 0.2, the author should explain the selection criteria of the variables
Fig3a: I don’t think the PCA has explained the variations between different soil sites very well, most of the data points were in the similar ellipse region
Author Response
Reviewer 1 comments
I have reviewed the manuscript entitled " Relationship of microbial activity with soil properties in banana plantations in Venezuela". The authors study relationship between microbial communities, soil physicochemical properties in different cropland area. The manuscript is generally well written. However, the some statistical analysis is problematic and the lack of details in the results section.
L70: should clarify which kind of microbial activity is discussed in the manuscript.
Authors: L70: Corrected. The following sentence was added: estimated indirectly by determining the basal respiration (O2 consumption in the medium or the concentration of CO2 released).
Fig 1b: raw data points and stdev is missing
Authors: Corrected. an appendix (Table A2) with the information of n, mean and stdev was added.
L195-197: the authors should do ANOVA or similar analysis to compare the nematode variables
Authors: Line 240. Corrected. The following sentence was added: …that no significant differences were found for both nematode variables (data not shown).
Authors: The Post Hoc Tests (Homogeneous Subsets were performed), whose results were not significant for the nematode variables for all the sites evaluated. An example of a sampling site (PL) is shown below:
|
PN |
||
|
Duncan |
||
|
|
N |
Subset for alpha = .05 |
|
NP |
1 |
|
|
L |
4 |
73,7 |
|
H |
4 |
92,95 |
|
Sig. |
|
0,164 |
|
Means for groups in homogeneous subsets are displayed. |
||
|
a Uses Harmonic Mean Sample Size = 4.000. |
||
|
FLN |
||
|
Duncan |
||
|
|
N |
Subset for alpha = .05 |
|
NP |
1 |
|
|
H |
4 |
7,125 |
|
L |
4 |
26,325 |
|
Sig. |
|
0,165 |
|
Means for groups in homogeneous subsets are displayed. |
||
|
a Uses Harmonic Mean Sample Size = 4.000. |
||
Fig 2: The logarithmic looks good but the author needs to show the stdev and number of replicates
Authors: Line 601. an appendix (Table A1) with the information of n, mean and stdev was added
L275: why the loading values have to be higher than 0.2, the author should explain the selection criteria of the variables
Authors: L289: Corrected. The following sentence was added: Only corresponding loadings greater than the absolute value of 0.2 are the best option for model interpretation and revealing the most important variables regarding the explanation of the response [41].
Citation: Yang, B.; Zhang, C.; Cheng, S.; Li, G.; Griebel, J.; Neuhaus, J. Novel Metabolic Signatures of Prostate Cancer Revealed by 1H-NMR Metabolomics of Urine. Diagnostics 2021, 11, 149. https://doi.org/10.3390/diagnostics1102014
Fig3a: I don’t think the PCA has explained the variations between different soil sites very well, most of the data points were in the similar ellipse region
Authors: We are agreed. L280: Corrected. The following sentence was added: PLS-DA could not distinguish high and low productivity levels of banana (Figure 3a). The first two main components (PC) explained 49.9% of the variables; however, no trends were detected in banana productivity differences

Reviewer 2 Report
The manuscript entitled “Relationship of microbial activity with soil properties in banana plantations in Venezuela” authored by Olivares et al., is a microbial diversity study in the banana fields. The reviewer gone through the manuscript and found that the manuscript is presented as per journal guidelines, the contents of the manuscript is in a defined format, and the table and figures are presented appropriately. The followings are the query and suggestions for improving the quality of the manuscript-
1. Please follow the same naming style for all authors
2. Delete the keywords “Free-living nematodes and soil” author should add “soil microbes” in the keywords
3. Briefly add the nutritional and economic value of the banana with a suitable citation
4. Rewrite “The most recent works related to the quality of …González-García et al. [12, 42 13].” For better clarity line no. 40-43
5. Delete “microbes” in line no. 46
6. Rewrite the sentence “Precisely these organisms constitute … soil aggregates stable [14,15]” line no. 47-53 for better clarity, it is better to write in small sentences
7. Table .1 is ok, it will be better to add the average yield in different farms
8. Method and material are described nicely
9. Figure 1 is ok
10. Check for “Figure 2a” it should be “Figure 2A” same with figure 2b and 3 also made correction
11. Figure 2 and 3 are ok
12. Table 2 and 3 are ok
13. Figure 4 and 5 are ok
14. The results and discussion section is written nicely and the interpretation of results in the discussion is ok.
15. The conclusion section should be modified for soundness
Overall, the manuscript is presented in a nice but there are many minor grammatical mistakes and the use of unscientific words which can be improved. The present study will help in the identification of beneficial soil microbial groups for banana yield improvement.
Author Response
The manuscript entitled “Relationship of microbial activity with soil properties in banana plantations in Venezuela” authored by Olivares et al., is a microbial diversity study in the banana fields. The reviewer gone through the manuscript and found that the manuscript is presented as per journal guidelines, the contents of the manuscript is in a defined format, and the table and figures are presented appropriately. The followings are the query and suggestions for improving the quality of the manuscript-
- Please follow the same naming style for all authors
Authors: Corrected.
- Delete the keywords “Free-living nematodes and soil” author should add “soil microbes” in the keywords
Authors: Corrected.
- Briefly add the nutritional and economic value of the banana with a suitable citation
Authors: L35. Corrected. The following sentence was added: Its composition highlights its richness in carbohydrates (20%), and an important source of Potassium, Vitamin C, and Iron.
- Rewrite “The most recent works related to the quality of …González-García et al. [12, 42 13].” For better clarity line no. 40-43
Authors: L35. Corrected. The following sentence was added: The most recent studies in Venezuela are those of
- Delete “microbes” in line no. 46
Authors: L46. Corrected.
- Rewrite the sentence “Precisely these organisms constitute … soil aggregates stable [14,15]” line no. 47-53 for better clarity, it is better to write in small sentences
Line 54: Corrected. The following sentence was added: These organisms are represented by unicellular and microscopic forms, nematodes, insect larvae, earthworms and arthropods, whose function is to transform all organic and inorganic compounds in the soil with direct or indirect influence on properties such as soil porosity, water transport and the union of particles for the formation of stable soil aggregates [14,15].
- Table .1 is ok, it will be better to add the average yield in different farms
Authors: Corrected.
- Method and material are described nicely
Authors: Thanks
- Figure 1 is ok
Authors: Thanks
- Check for “Figure 2a” it should be “Figure 2A” same with figure 2b and 3 also made correction
Authors: Corrected.
- Figure 2 and 3 are ok
Authors: Thanks
- Table 2 and 3 are ok
Authors: Thanks
- Figure 4 and 5 are ok
Authors: Thanks
- The results and discussion section is written nicely and the interpretation of results in the discussion is ok.
Authors: Thanks
- The conclusion section should be modified for soundness
Authors: Corrected.
Overall, the manuscript is presented in a nice but there are many minor grammatical mistakes and the use of unscientific words which can be improved. The present study will help in the identification of beneficial soil microbial groups for banana yield improvement.
Authors: Thanks
Reviewer 3 Report
This manuscript submitted to ‘Sustainability’ deals with relationship of microbial activity with soil properties in banana plantations in Venezuela. The authors succeeded in demonstrating how soil properties influence the production of a complex and variable system there. It harnessed machine learning algorithms as Debiased Sparse Partial Correlation (DSPC), Partial Least Squares - Discriminant Analysis (PLS-DA) and Random Forest to explore a variety of variables in 6 agricultural fields located in two of the main banana production areas of Venezuela. The research article is full of many relevant references and statistical analyzes that are supportive and useful in analyzing the results. Yet, I do NOT understand why they did not standardize their sampling procedures (e.g., number of samples) across the surveyed fields, although the study considered the same number of undisturbed samples (36) obtained with an Uhland type sampler (mean depth 29.6 ± 17.6 cm) to obtain the saturated hydraulic conductivity (Ks, cm.h-1). The presented figures are generally OK but some needs to have more explanation especially for abbreviation, e.g. Figure 3. needs more clarification for the rest of symbols there. Moreover, the writing style is somewhat awkward with some misprints, grammar faults, and construction of some sentences in this manuscript, to name but few:
1) “For determining the relative importance of different order and genre nematodes” Should be ““For determining the relative importance of different nematode orders and genera”.
2) “with 4 four replicated” 4 is repeated.
3) “In studies recent, no significant…” Should be “In recent studies, no significant…”
4) respiration (r=1.00; p= 0.014), and NO3 (r= 1.00; p= 0.032). I wonder how the correlation coefficient reached its maximum point with such relatively modest statistical probability level.
5) “but in soil science as well. general.” The word “general” is alone !!
6) “and employment to populations rural order/genre in banana” misprints and grammar faults.
7) According to results, phytonematodes predominate in the banana system evaluated with ? (were) eleven important genera (Figure 1a), with Rotylenchulus
Therefore, I would suggest rewriting the manuscript and resubmitting it again.
Author Response
This manuscript submitted to ‘Sustainability’ deals with relationship of microbial activity with soil properties in banana plantations in Venezuela. The authors succeeded in demonstrating how soil properties influence the production of a complex and variable system there. It harnessed machine learning algorithms as Debiased Sparse Partial Correlation (DSPC), Partial Least Squares - Discriminant Analysis (PLS-DA) and Random Forest to explore a variety of variables in 6 agricultural fields located in two of the main banana production areas of Venezuela. The research article is full of many relevant references and statistical analyzes that are supportive and useful in analyzing the results.
Yet, I do NOT understand why they did not standardize their sampling procedures (e.g., number of samples) across the surveyed fields, although the study considered the same number of undisturbed samples (36) obtained with an Uhland type sampler (mean depth 29.6 ± 17.6 cm) to obtain the saturated hydraulic conductivity (Ks, cm.h-1).
Authors: Soil and banana productivity samplings were carried out according to the guidelines of Rosales et al. 2008. In that document, a group of international banana experts established the following:
Each site (high and low productivity) will be divided into sampling areas of approximately 1 hectare; the number will depend on the size of the farm. If the farms to be studied are smaller than 10 hectares, it is recommended to only take a sampling area in the "high" site and one in the "low productivity" site. On farms of 10 to 25 hectares, two sampling areas per site are recommended. If the farm has between 26 and 50 hectares, 3 areas per "high" site and 3 per "low productivity" site would be made. Farms with more than 50 hectares would be diagnosed with 4 sampling points in high production sites and 4 in low production or vigor sites.
- Rosales, F.E.; Pocasangre, L.E.; Trejos, J.; Serrano, E.; Peña, W. Guía de diagnóstico de la calidad y salud de suelos bananeros. Bioversity International: Roma. Italy. 2008; pp. 48-80. https://www.bioversityinternational.org/fileadmin/user_upload/online_library/publications/pdfs/1375.pdf
The presented figures are generally OK but some needs to have more explanation especially for abbreviation, e.g. Figure 3. needs more clarification for the rest of symbols there.
Authors: Line 427. The following sentence was added: Figure 3. A) Scores plot between the Component 1 and Component 2 of the PLS-DA. Note: Data code composed of three digits, the first digit is the site (PL, KA, BA, SM, PA, and CH), followed by the level of productivity: (H) High and (L) low, and finally the repetition number (1-4).
Moreover, the writing style is somewhat awkward with some misprints, grammar faults, and construction of some sentences in this manuscript, to name but few:
1) “For determining the relative importance of different order and genre nematodes” Should be ““For determining the relative importance of different nematode orders and genera”.
Authors: Line 199: corrected.
2) “with 4 four replicated” 4 is repeated.
Authors: Line 149. corrected.
3) “In studies recent, no significant…” Should be “In recent studies, no significant…”
Authors: Line 525: corrected.
4) respiration (r=1.00; p= 0.014), and NO3 (r= 1.00; p= 0.032). I wonder how the correlation coefficient reached its maximum point with such relatively modest statistical probability level.
Authors: Line 532: Here we present the results of the deviant sparse partial correlation (DSPC) algorithm that is based on the recently proposed deparsified graph lasso modeling procedure (Jankova, 2015). A key assumption underlying our modeling strategy is that the number of true connections between variables is much smaller than the available sample size, that is, the true network of partial correlations between variables is sparse.
This assumption is strongly supported by both empirical evidence and theoretical calculations (Gardner et al., 2003; Jeong et al., 2001; Leclerc, 2008). Under this assumption, DSPC reconstructs a graphical model and provides partial correlation coefficients and P-values ​​for each pair of variables in the data set. Therefore, DSPC allows to discover the connectivity between many variables using fewer samples. The results can be visualized as weighted networks where the nodes represent the variables, and the edges represent partial correlation coefficients or the associated P-values.
5) “but in soil science as well. general.” The word “general” is alone !!
Authors: Line 90: corrected.
6) “and employment to populations rural order/genre in banana” misprints and grammar faults.
Authors: Line 37: corrected. the phrase was deleted
Line 233: order/genre was deleted
7) According to results, phytonematodes predominate in the banana system evaluated with ? (were) eleven important genera (Figure 1a), with Rotylenchulus
Therefore, I would suggest rewriting the manuscript and resubmitting it again.
Authors: Line 280. corrected. According to results, Phytonematodes predominate in the banana system being eleven important genera (Figure 1A), like Rotylenchulus
Authors: Corrections to grammatical errors in the manuscript were made
Reviewer 4 Report
Only some comments are included in the ms.
In general, the ms is valuable.

Author Response
In general, the ms is valuable.
Authors: Thanks
Line 18: delete and
Authors: Corrected.
Citation (1) I am not sure if this ms is the most appropriate to take the general overview of the banana plantation in the world. The article is related with banana grow in Venezuela.
Authors: Corrected. added this new reference:
Martínez-Solórzano, G.E.; Rey-Brina, J.C. Bananos (Musa AAA): Importance, production, and trade in Covid-19 ti-mes. Agronomía Mesoamericana 2021, 32(3), 1034-1046. http://dx.doi.org/10.15517/am.v32i3.43610
Round 2
Reviewer 1 Report
the revised version looks good